# Ageing and Polypharmacy in Mesenchymal Stromal Cells: Metabolic Impact Assessed by Hyperspectral Imaging of Autofluorescence

**DOI:** 10.3390/ijms25115830

**Published:** 2024-05-27

**Authors:** Chandrasekara M. N. Chandrasekara, Gizem Gemikonakli, John Mach, Rui Sang, Ayad G. Anwer, Adnan Agha, Ewa M. Goldys, Sarah N. Hilmer, Jared M. Campbell

**Affiliations:** 1Graduate School of Biomedical Engineering, Faculty of Engineering, University of New South Wales, Sydney, NSW 2052, Australia; navinduchandrasekara@outlook.com (C.M.N.C.); r.sang@unsw.edu.au (R.S.); a.anwer@unsw.edu.au (A.G.A.); a.agha@student.unsw.edu.au (A.A.); e.goldys@unsw.edu.au (E.M.G.); 2Laboratory of Ageing and Pharmacology, Kolling Institute, Northern Sydney Local Health District and Faculty of Medicine and Health, The University of Sydney, Camperdown, NSW 2050, Australia; ggem6186@uni.sydney.edu.au (G.G.); john.mach@sydney.edu.au (J.M.); sarah.hilmer@sydney.edu.au (S.N.H.)

**Keywords:** mesenchymal stem cell, aging, autofluorescence, non-invasive imaging, spectral microscopy, polypharmacy

## Abstract

The impact of age on mesenchymal stromal cell (MSC) characteristics has been well researched. However, increased age is concomitant with increased prevalence of polypharmacy. This adjustable factor may have further implications for the functionality of MSCs and the effectiveness of autologous MSC procedures. We applied hyperspectral microscopy of cell autofluorescence—a non-invasive imaging technique used to characterise cytometabolic heterogeneity—to identify changes in the autofluorescence signals of MSCs from (1) young mice, (2) old mice, (3) young mice randomised to receive polypharmacy (9–10 weeks of oral therapeutic doses of simvastatin, metoprolol, oxycodone, oxybutynin and citalopram), and (4) old mice randomised to receive polypharmacy. Principal Component Analysis and Logistic Regression Analysis were used to assess alterations in spectral and associated metabolic characteristics. Modelling demonstrated that cells from young mice receiving polypharmacy had less NAD(P)H and increased porphyrin relative to cells from old control mice, allowing for effective separation of the two groups (AUC of ROC curve > 0.94). Similarly, cells from old polypharmacy mice were accurately separated from those from young controls due to lower levels of NAD(P)H (*p* < 0.001) and higher porphyrin (*p* < 0.001), allowing for an extremely accurate logistic regression (AUC of ROC curve = 0.99). This polypharmacy regimen may have a more profound impact on MSCs than ageing, and can simultaneously reduce optical redox ratio (ORR) and increase porphyrin levels. This has implications for the use of autologous MSCs for older patients with chronic disease.

## 1. Introduction

In old age, the increase in multimorbidity results in a concomitant increase in the number of medications consumed on a regular basis [1]. Polypharmacy refers to a patient’s use of multiple medications, with the most common definition being the daily use of at least five medications [2]. Within Australia, the number of people affected by polypharmacy increased by 52% from 543,950 to 828,950 in the period from 2006 to 2017, with those aged 80–89 experiencing the highest rates of polypharmacy [3]. Although the harmful effects of polypharmacy on human health are difficult to investigate experimentally, observational studies suggesting associations with increased falls, frailty, cognitive impairment and adverse geriatric outcomes [4,5,6,7] are supported by experimental findings in mice [8,9,10].

Mesenchymal stromal cell (MSC) transplantation is a regenerative medicine therapy with the potential to contribute to numerous areas of human disease. Autologous transplant is generally desirable as it avoids the issue of immuno-incompatibility. Examples where MSC therapy has shown promise include diabetes [11], heart disease [12], and autoimmune diseases [13]. All these examples are major contributors to requiring polypharmacy. As such, MSC therapy has significant potential to ameliorate the burden of polypharmacy by offering a non-pharmacological intervention for the treatment of these diseases. However, as recipients of MSC therapy consequently have a high likelihood of being exposed to polypharmacy at the time of autologous MSC collection, the unknown impact of polypharmacy on MSC characteristics requires study. Moreover, similar to polypharmacy, people receiving MSC therapy are likely to be of increased age, which has been well established to negatively impact MSC quality [14,15]. As such, this investigation was undertaken to determine the interaction of age and polypharmacy on MSC biology.

Five medications—simvastatin, metoprolol, oxybutynin, oxycodone, and citalopram—were used in this study to model those commonly consumed in combination by older people [16]: simvastatin is a lipid-lowering medication used to treat high serum cholesterol [17]; metoprolol is a β-blocker used to treat high blood pressure and cardiovascular disease [18]; oxybutynin is prescribed to treat symptoms of an overactive bladder [19]; oxycodone is an opioid used to treat moderate to severe pain [20]; and citalopram is a selective serotonin reuptake inhibitor used to treat depression [21]. To model young adult and old age, mice were 6.5 months and 25.5 months old at the time of euthanasia following 9–10 weeks of treatment.

Numerous intracellular molecules have autofluorescent properties, with identifiable excitation and emission spectra. Contributing fluorophores include the metabolic coenzymes NAD(P)H and flavins, structural collagen family proteins, and heme proteins such cytochrome c and protoporphyrin IX [22,23]. As such, in this project, we have applied a multispectral approach to studying the effect of in vitro ageing and polypharmacy on the redox state of MSCs. This technology uses a modified fluorescence microscope with an extended number of potential spectral channels (20–100) [24] to repeatedly image a single field of view at selected excitation wavelengths, capturing native emissions in specific wavelength ranges. The resultant dataset provides rich cellular details and quantitative information based on cell autofluorescence, which is a direct indicator of intracellular state and activities, that we have used to study ageing in various systems [14,25]. Of greatest relevance was that we demonstrated the existence of a spectral signature for advancing in vitro age in human MSCs, which could be applied across lines and confirmed the contribution of an altered redox balance—indicated by a shift in the ratio of NAD(P)H to flavins—to MSC ageing [14]. As well as giving information on the abundance of specific native fluorophores, the broad-spectrum approach contains rich biomarker information which can be mapped to further cellular conditions, including ROS levels [26], cell cycle position [24] and ploidy [27]. Due to the generally exacerbating effect of polypharmacy on ageing (e.g., increased frailty and cognitive decline [10,28]), we hypothesise that MSCs from older mice exposed to polypharmacy will have a more severe ageing phenotype—i.e., lower ratio of NAD(P)H to FAD compared to MSCs from mice that have not been exposed to polypharmacy, and a more clearly distinct spectral profile compared to MSCs from young, untreated mice. As polypharmacy does not have as strong an effect on younger organisms compared to old [10,29], we hypothesise that the effect on younger mice will be more muted.

## 2. Results

All experiments were performed on MSCs that had been cultured to passage 2. The old mice were aged 25.5 months at the time of sacrifice and cell line derivation, while the young mice were aged 6.5 months. For MSC cell surface markers, the lines co expressed positive markers CD105, CD29, SCA-1 in >80% of cells and the negative marker CD45 in <3% of cells (data not shown).

### 2.1. Cells from Young and Old Control Mice

The results showed a reduction in NAD(P)H in MSCs from old mice compared to young (Channel 11, Figure 1a, *p* < 0.001). The secondary channels used to produce a model between YC and OC MSCs included the product of Flavin Channels and ORR Channels. This provides the necessary labelled training data to produce a relatively strong model, as highlighted by the ROC curve (Figure 1g), which generated an AUC of 0.92.

### 2.2. Cells from Young Polypharmacy-Treated and Old Polypharmacy-Treated Mice

There were no statistically significant channels when comparing cells from young polypharmacy-treated (YP) and old polypharmacy-treated (OP) mice. Plotting the three channels with the smallest *p*-values yielded little success as the cells from each group did not separate along these axes (Figure 2a) and the ROC AUC was just 0.83 (Figure 2d).

### 2.3. Cells from Young Control and Young Polypharmacy-Treated Mice

There was significantly less NAD(P)H present in MSCs from mice that had been exposed to polypharmacy (Figure 3a), despite the cells being the same age. As the NAD(P)H channels consistently showed a drop off between the young control (YC) and YP cells, and very few flavin channels showed any significant differences, the ORR also followed the same relationship between the two groups (Figure 3c). Porphyrin levels were also significantly greater in the cells from mice receiving polypharmacy, as shown by the comparison of Porphyrin Channel 2 (Figure 3b), which is the product of Channel 8 and Channel 9—the most statistically significant porphyrin channels. These relationships enabled strong separation of the two groups (Figure 3d,g AUC = 0.91).

### 2.4. Cells from Old Control and Old Polypharmacy-Treated Mice

All the porphyrin channels again showed significant differences when comparing control and polypharmacy cells from the same age group. The only secondary channels used to create the model for this comparison were the product of porphyrin channels. This model an excellent ROC curve (Figure 4g) with a high AUC (0.95). Plotting the 3 most significant primary channels—Channel 8, Channel 26, and Channel 28—produced good separation between the two datasets (Figure 4d).

### 2.5. Cells from Young Polypharmacy-Treated and Old Control Mice

The porphyrin and redox channels were successful in separating old control (OC) and YP cells along their respective axes (Figure 5d). YP cells had higher mean porphyrin, and lower ORR (Figure 5a–c); this allowed for secondary channels to be created using the product of porphyrin channels, and ORR to act as training data for LRA. Thus, the subsequent model created by logistic regression analysis (LRA) (Figure 5f) was accurate in its classifications of YP and OC cells based on significant primary and secondary channels. The area under the ROC curve was 0.94 (Figure 5g), highlighting the accuracy at which the training data can produce a model that correctly categorises data.

### 2.6. Cells from Young Control and Old Polypharmacy-Treated Mice

Separation between YC and OP cells was achieved using porphyrin, redox, and NAD(P)H channels, with PCA (Figure 6e), LRA (Figure 6f), and simple 3D plots (Figure 6d) all showing clear separation (Figure 6 g; ROC AUC = 0.99). For cells from old mice receiving polypharmacy, the boxplots show significant decreases in NAD(P)H levels (Figure 6a) and ORR (Figure 6c) and increases in porphyrin levels (Figure 6b).

## 3. Discussion

Comparisons between pairs of cell groups allow for an understanding of the individual and joint consequences of ageing and polypharmacy on MSCs. This has implications for MSC lines derived for autologous therapy, where the self-donors have a strong likelihood of being both old and exposed to polypharmacy. As hypothesised, polypharmacy caused a reduction in metabolic activity, measured using the ORR. Porphyrin channels showed significant differences between control groups and mice receiving polypharmacy, with cells from polypharmacy mice expressing greater levels of PPIX (a common porphyrin found in animals, especially MSCs [30,31]). This increase in PPIX is a possible contributor to the functional impairments observed in the mice used for this study as PPIX is linked to sarcopenia (a condition describing the loss of muscle function and muscle mass) [16,32]. The relationships found between young cells and old cells corroborates the findings from previous studies: NAD(P)H levels, and therefore the ORR, decrease with age, causing MSCs to proliferate less readily [14]. Young and old control cells can be accurately separated through a modelling process such as LRA by training the data using NAD(P)H channels and ORR channels. However, when comparing young and old cells treated with polypharmacy, there were no significant differences between channels. This implies that the effects of polypharmacy on autofluorophores is more impactful than that of ageing and is masking the chemical changes usually observed during the ageing process.

Two relationships appear in the comparison of control cells to those that have experienced polypharmacy; NAD(P)H levels decrease when cells are exposed to polypharmacy, and porphyrin levels increase. The hypothesised relationship between NAD(P)H and polypharmacy was confirmed in this study, as the ORR decrease was lower in YP cells compared to YC cells. Thus, like OC MSCs, the YP MSCs are expected to divide more slowly [30,33], reducing their suitability to be applied in autologous stem cell therapies. The increase in porphyrin was not expected and is a greater cause for concern as it is a marker for a variety of diseases [32,34]. Since the models created to separate cells from control mice and polypharmacy mice primarily used porphyrin channels, porphyrin can be considered the predominant differentiator in this comparison.

This leads to the comparison of YP cells with OC cells, and YC cells with OP cells. Since ageing and polypharmacy reduce NAD(P)H levels in cells and hence reduce the ORR [33], it was expected that these YP and OC cells would have similar characteristics and would be difficult to distinguish from each other. However, it was found that YP cells had lower NAD(P)H and ORR, whilst having greater levels of porphyrin, allowing the two groups to be separated accurately using LRA (AUC of ROC curve 0.94). These differences between YP and OC cells explain the inability to differentiate between YP and OP cells in the previous comparison, as it appears polypharmacy masks the effects of ageing by lowering NAD(P)H and ORR far more than the ageing process and increasing porphyrin levels in all cells afflicted by polypharmacy. Finally, the model for separating YC and OP cells is extremely accurate (ROC AUC = 0.99) as the adverse effects of polypharmacy are compounded by the effects of ageing, causing OP MSCs to have much lower NAD(P)H and ORR than YC cells, and displaying stronger autofluorescence signals when observing porphyrin channels.

This study was designed to investigate the effect of polypharmacy—the exposure of an organism to multiple medications and the health outcomes thereof—as opposed to the specific impacts of the individual medications on subsequently derived MSCs. As such, the individual impacts of the applied factors (metoprolol, citalopram, simvastatin, oxycodone, oxybutynin) on stem cell health need to be considered. Previously, the treatment of spontaneously hypertensive rats with the anti-hypertensive agent with metoprolol reduced senescence as well as the levels of ROS in subsequently isolated cardiac stem cells [35]. These cell lines also had higher migration, proliferation and stemness than cells from rats with untreated hypertension. Direct treatment of MSCs with the selective serotonin reuptake inhibitor citalopram resulted in enhanced neuronal characteristics on directed differentiation, as well as decreased cell death and increased population doublings [36]. A study on canine MSCs found that simvastatin—which blocks the mevalonate pathways responsible for the production of cholesterol—enhanced the proliferation and caused upregulated gene expression of the cell cycle regulators *Cyclin D1* and *Cyclin D2*, the proliferation marker *Ki-67*, the anti-apoptotic gene *Bcl-1* and the pluripotency markers *Rex1* and *Oct4* when administered to in vitro cultured canine BM-MSCS at low doses (0.1 to 1 nm), although higher doses (up to 100 nm) had a negative effect [37]. Another study found that simvastatin reduced the in vitro proliferation of human MSCs; however, it increased the percentage of the subset of smaller MSCs that were proliferating [38]. Given the tendency for MSCs to be affected by cellular hypertrophy over long-term in vitro culture [14], this was interpreted by the authors as being suggestive of a protective impact on MSC health. No studies were found that investigated the likely impact of oxycodone or oxybutynin on stem cells characteristics. As the majority of medications have been found, in isolation, to have a generally positive impact on MSC characteristics, we conclude that the ageing-like effects found in this work can be attributed to the impacts of polypharmacy on exposed mice.

The use of hyperspectral microscopy in this study offers many advantages over other imaging methods as it is non-destructive and does not require the use of stains—practices that have the potential to influence the chemical composition, and hence, the data collected from the cells. Furthermore, the cells used in this study provide an excellent basis for the study of the effects of polypharmacy since the mice were treated with the five medications and cells were then taken from the mice, rather than applying the polypharmacy exposure to the cells in vitro. However, this model only studies the effects of polypharmacy in the absence of disease. Older people usually have effects of polypharmacy plus the effects of diseases for which the medications are prescribed. This study is also generalized to the medications investigated. Future studies should investigate other common combinations of polypharmacy and probe the reversibility of these effects via deprescribing, the planned process of ceasing medications with clinical guidance to manage polypharmacy. Finally, the impact of polypharmacy on health is known to be affected by sex [8], and as such, different results could be obtained in a study which included or focused on male mice.

Our findings support that the biochemistry of MSC lines is influenced by the status and exposures experienced by their original donor. MSC lines used for autologous therapies—with the exception of pre-banked cells—invariably come from individuals experiencing at least one pathology and who have a high likelihood of polypharmacy and advanced age. This warrants serious consideration in the design of studies and the implementation of therapies. In particular, it may be a contributing factor to the generally heterogenous nature and behaviour of human MSC lines [39] as well as the difficulties in translating the findings of pre-clinical studies to patient outcomes [40,41]. Further studies demonstrating effects on clinical outcomes are needed to make any recommendations; however, de-prescribing—the careful and considered withdrawal of medications whose benefits may no longer outweigh their negative effects—is already good clinical practice [42]. This would represent a new angle of research for ‘pre’-habilitation—an emerging health care strategy with applications including preparation for stem cell therapies [43,44]—where efforts are made to improve the patient’s fitness so that they can better bear and recover from intervention as in the proposed scenario, the patient’s health would be improved with the goal of improving the quality of the therapy itself. In addition to deprescribing, interventions like nutritional supplementation with the metabolic precursor NMN—which has been shown to be able to improve the quality of subsequently derived MSC lines [45]—may also be able to improve the quality of MSC lines from older people exposed to polypharmacy undergoing autologous stem cell therapy.

### Conclusions

With polypharmacy rates among older people increasing and the ageing population a growing global concern, it is vital that medical procedures involving MSCs are still effective in patients whose cells have been affected by ageing, disease, and polypharmacy. This project was able to develop models to categorise cells based on their autofluorescence spectra and reports novel findings on the effects of polypharmacy and ageing on MSCs. Polypharmacy was found to have a greater influence on the biochemical composition of cells than ageing as it concurrently reduced cellular metabolic activity to a greater extent and increased expression of porphyrin, a marker for a variety of human diseases and medical conditions. This suggests that autologous stem cell transplants are likely to be less effective in elderly patients. Further research is required to confirm the clinical impact on MSC therapy and potential strategies to mitigate the effect.

## 4. Materials and Methods

### 4.1. Animal Care and Cell Culture

All experiments were carried out according to the guidelines of the National Health and Medical Research Council of Australia and approved by the Animal Ethics Committee of the Northern Sydney Local Health District, Sydney, Australia (RESP/16/348). Mesenchymal stem cells were collected from young (approximately 6.5 months) and old (approximately 25.5 months), female C57BL/6 mice that had been randomized to receive control or polypharmacy exposures for 9–10 weeks before euthanasia. Animal care and treatment for this cohort have been described previously [16]. At the point of analysis, after attrition due to premature mortality amongst the aged mice and unsuccessful line derivation, the groups contained 6, 6, 6 and 4 mice for the young control, old control, young polypharmacy and old polypharmacy groups, respectively. Data within groups was pooled and analysed at the single cell level, as detailed in Section 4.4. Up to 5 animals were housed per cage and kept on a 12 h light–dark cycle, with food and water provided ad libitum. The polypharmacy treatment comprised drug classes that are commonly prescribed in older people, have similar pharmacokinetics and pharmacodynamic characteristics in humans and mice and are not toxic when given to healthy mice in monotherapy. The drugs were metoprolol (350 mg/kg/day), simvastatin (20 mg/kg/day), oxycodone and oxybutynin at their minimum effective doses (5 and 27.2 mg/kg/day) and citalopram at 150% the minimum dose (15 mg/kg/day). The polypharmacy intervention was given ad libitum via chow and water as described previously [9].

Following euthanasia, the hind limbs were dissected and MSC lines were derived by flushing the tibia and femur with MSC culture media (α-MEM with sodium bicarbonate, without L-glutamine, ribonucleosides and deoxyribonucleosides supplemented with 10% fetal bovine serum, sodium pyruvate, 2 mM L-glutamine, 100 μM L-ascorbic acid, 50 U/mL penicillin and 50 μg/mL streptomycin) and culturing the bone marrow in cell culture dishes at 5% O_2_, 6% CO_2_ and a balance of N_2_ at 37 °C according to the protocol described in [46]. The cells were passaged twice to reduce the presence of macrophages, blood cells and fat. The presence of colony forming units was confirmed by plating at low density (Appendix A). MSC cell surface markers were checked using the Mouse MSC 4-color flow cytometry kit for CD105, CD29, Sca-1, CD45 (R&D systems, FMC003, Appendix A). The cells were then plated in 35 mm polymer coverslip imaging dishes (Cell E&G PDP00002-20) for imaging on the hyperspectral microscope after 48 h.

### 4.2. Cell Groups and Study Design

Comparisons were performed between 4 cell groups: Young Control (YC), Old Control (OC), Young Polypharmacy-treated (YP), and Old Polypharmacy-treated (OP). This yielded 6 combinations which provided information regarding the effects of ageing, polypharmacy, and a combination of the two factors on MSC biology (Table 1).

### 4.3. Hyperspectral Imaging

Immediately prior to imaging, the media was replaced with spectrally neutral Hank’s balanced salt solution. Hyperspectral microscopy was performed on an Olympus IX83 microscope (Olympus, Japan, Tokyo) with a NuVu electron multiplying charge coupling device camera, and a 40× immersion oil objective lens. Excitation was achieved via low powered LEDs (wavelengths detailed in Table 2), and emission wavelengths were isolated by optical filters centred at 414, 451, 575, and 594 nm wavelengths (±20 nm) and a long pass at 675 nm. The pairs of excitation/emission wavelength ranges formed the hyperspectral channels used in this study, and we selected 31 channels, as shown in Table 2. Many of these channels closely correspond to excitation and emission profiles of fluorophores [47] present within cells (Table 2). Channel 32 is a brightfield channel used in the creation of cellular masks. Image pre-processing was undertaken as detailed previously, and the images were flattened by removing background autofluorescence signals and using a calibration fluid with a known concentration of NADH and FAD [26,48]. Image smoothing was achieved through the application of a wavelet filter, assuming that noise in the image originated from shot noise. Cellular masks were defined manually prior to analysis.

### 4.4. Image Analysis

Average fluorescence intensity values for each cell were calculated for each of the investigated channels. The collection of such values for each cell in the cell group under investigation in a particular channel represents a basic unit in our analysis. The *t*-test was used to test for statistically significant differences (e.g., between cell groups on a channel-by-channel basis) (the code used to calculate *p*-values is provided in Appendix A). The Pearson correlation coefficient (r) (for average fluorescence signal for each cell in a specific group) between pairs of channels (Appendix A) was used to assess linear correlation between channels, with $\left|r\right|<0.3$ indicating low correlation. Optical redox ratio (ORR) and secondary channels were calculated as described in Appendix A). Significant channels showing statistically significant differences for cell groups under investigation (*p* < 0.05) were noted for further analysis (non-significant findings are reported in Appendix A), including assessment of Principal component (PC) scores (Appendix A).

## Figures and Tables

**Figure 1 ijms-25-05830-f001:**
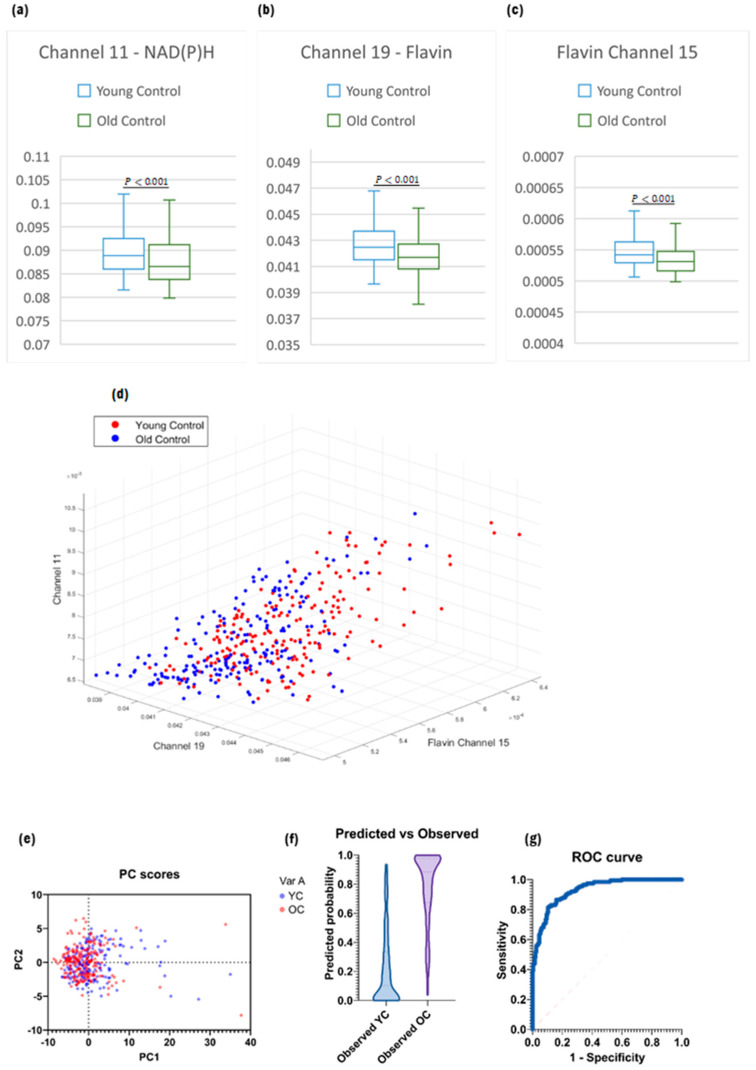
(**a**) Boxplot of Channel 11, based on two sample *t*-test for independent means. (**b**) Boxplot of Channel 19. (**c**) Boxplot of Channel 15 from Flavin secondary channels. (**d**) 3D plot of the datasets based on the three strongest primary channels. (**e**) Principal Component (PC) scores with the strongest loading from PCA analysis. (**f**) Predicted vs. observed distributions of the model generated by logistic regression analysis. (**g**) ROC curve of the model.

**Figure 2 ijms-25-05830-f002:**
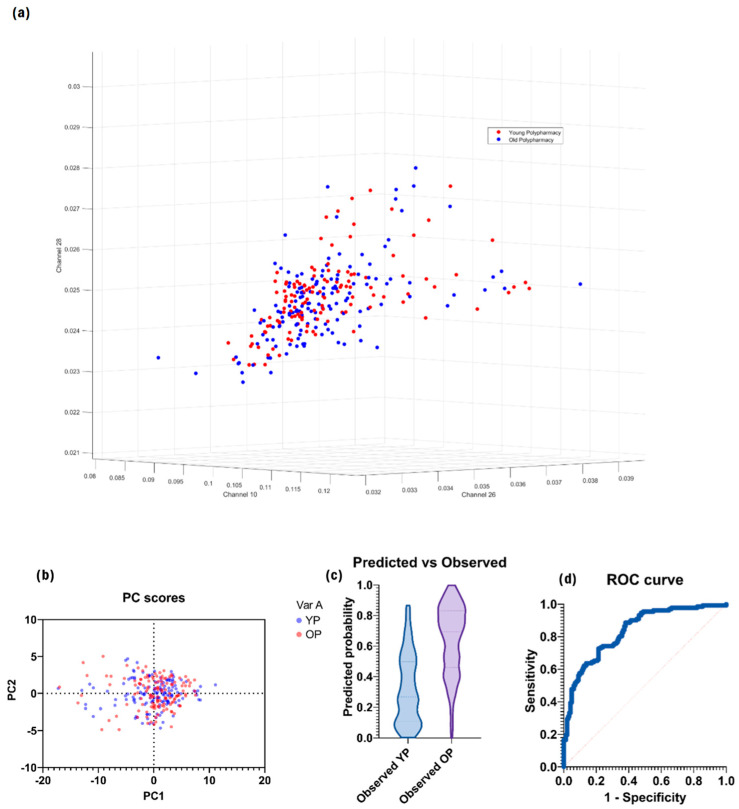
(**a**) Scatter plot with little separation between the datasets. (**b**) PC scores with little separation between datasets based on the three strongest primary channels. (**c**) Predicted vs. observed distributions of the model generated by logistic regression analysis. (**d**) ROC curve of the model with poor separation between datasets.

**Figure 3 ijms-25-05830-f003:**
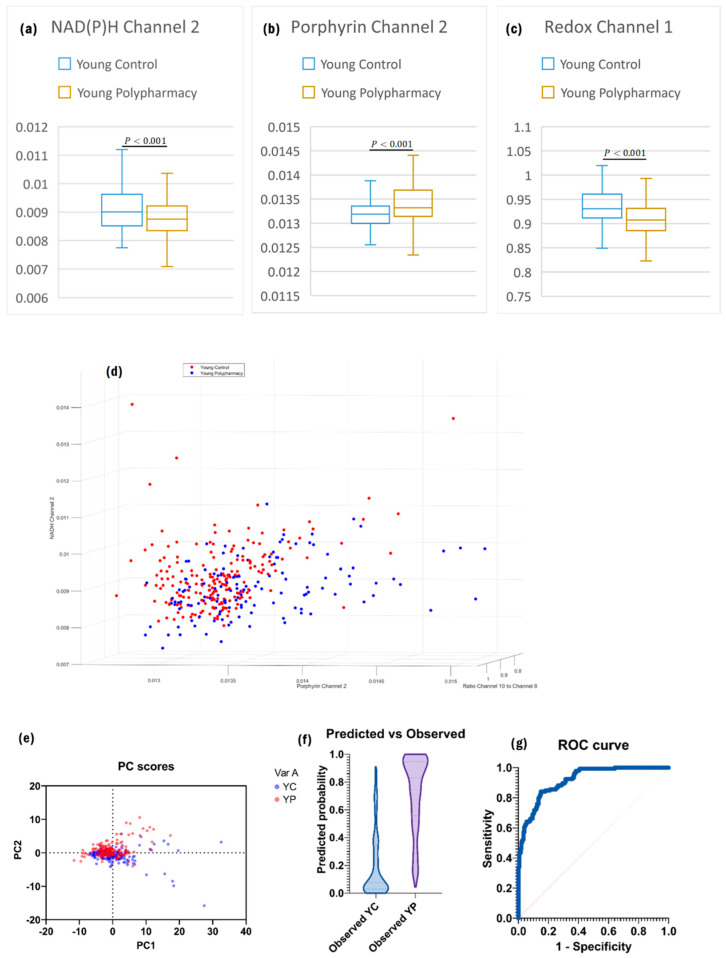
(**a**) Boxplot of Channel 2 from NAD(P)H secondary channels, based on two sample *t*-test for independent means. (**b**) Boxplot of Channel 2 from Porphyrin secondary channels. (**c**) Boxplot of Channel 1 from Redox Ratio Channels. (**d**) 3D plot of the datasets based on the three strongest primary channels. (**e**) PC scores with the strongest loading from PCA analysis. (**f**) Predicted vs. observed distributions of the model generated by logistic regression analysis. (**g**) ROC curve of the model showing good accuracy.

**Figure 4 ijms-25-05830-f004:**
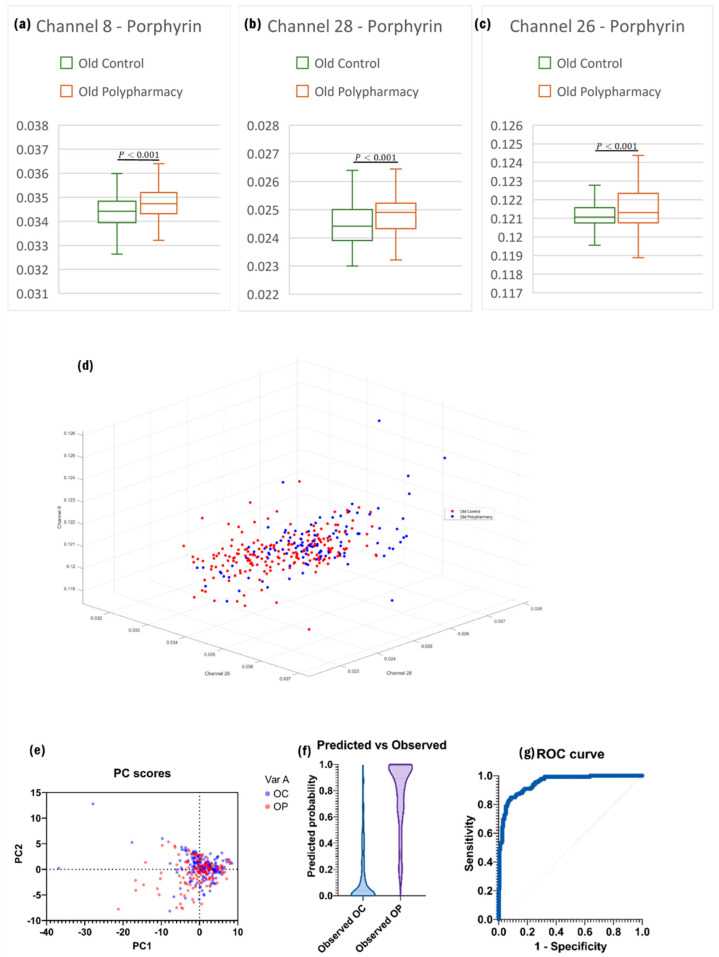
(**a**) Boxplot of Channel 8, based on two sample *t*-test for independent means. (**b**) Boxplot of Channel 28 Channels. (**c**) Boxplot of Channel 26. (**d**) 3D plot of the datasets based on the three strongest primary channels. (**e**) PC scores with the strongest loading from PCA analysis. (**f**) Predicted vs. observed distributions of the model generated by logistic regression analysis. (**g**) ROC curve of the model showing great accuracy.

**Figure 5 ijms-25-05830-f005:**
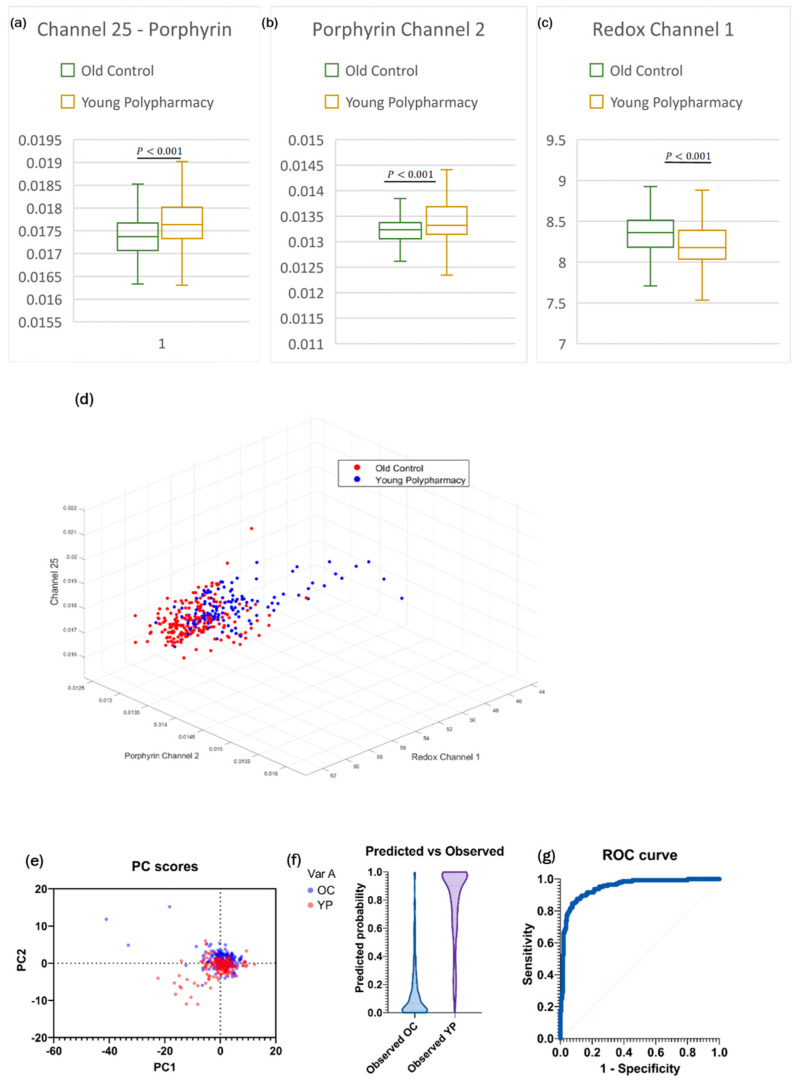
(**a**) Boxplot of Channel 25, based on two sample *t*-test for independent means. (**b**) Boxplot of Channel 2 from Porphyrin secondary channels. (**c**) Boxplot of Channel 1 from Redox Ratio Channels. (**d**) 3D plot of the datasets based on the three strongest primary channels. (**e**) PC score with the strongest loading from PCA analysis. (**f**) Predicted vs. observed distributions of the model generated by logistic regression analysis. (**g**) ROC curve of the model showing excellent accuracy.

**Figure 6 ijms-25-05830-f006:**
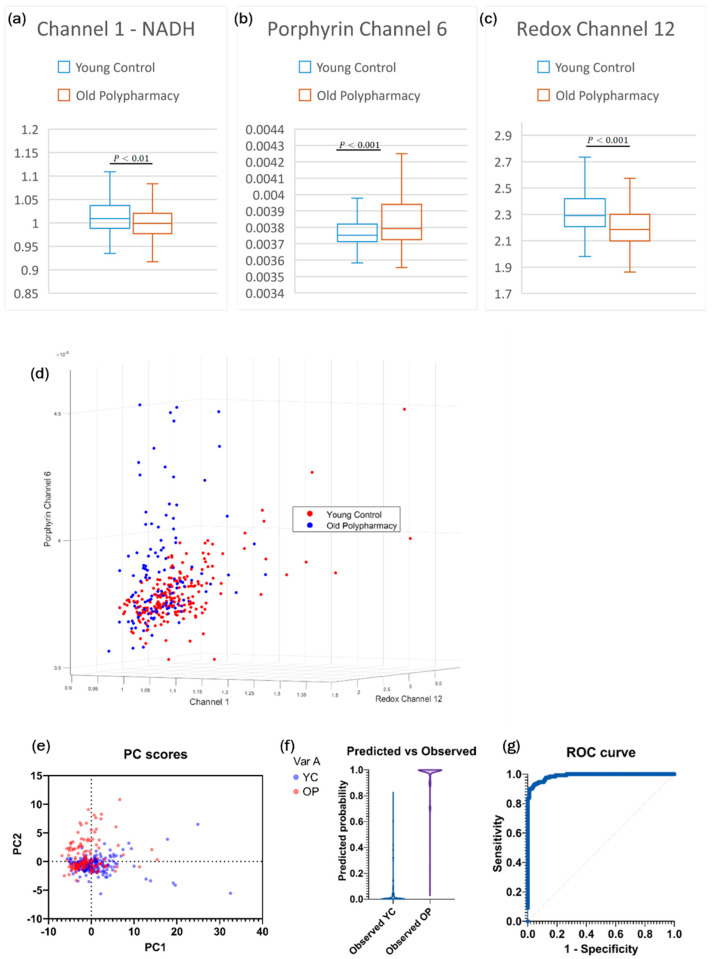
(**a**) Boxplot of Channel 1, based on two sample *t*-test for independent means. (**b**) Boxplot of Channel 6 from Porphyrin secondary channels. (**c**) Boxplot of Channel 12 from Redox Ratio Channels. (**d**) 3D plot of the datasets based on the three strongest primary channels. (**e**) PC scores with the strongest loading from PCA analysis. (**f**) Predicted vs. observed distributions of the model generated by logistic regression analysis. (**g**) ROC curve of the model showing near perfect accuracy with an AUC of 0.99.

**Table 1 ijms-25-05830-t001:** Cell group comparisons.

Pairing of Cell Groups	Information from Comparison
Young Control vs. Old Control	Identify changes in autofluorescence based on age and compare this to previous studies on the effects of ageing on MSC autofluorescence to ensure results are consistent.
Young Polypharmacy vs. Old Polypharmacy	Controlling for polypharmacy effects, identify changes in autofluorescence based on age.
Young Control vs. Young Polypharmacy	Identify differential changes in autofluorescence based on the effects of polypharmacy in young age.
Old Control vs. Old Polypharmacy	Identify differential changes in autofluorescence based on the effects of polypharmacy in old age.
Young Polypharmacy vs. Old Control	Compare the impacts of polypharmacy in young age and old age without polypharmacy.
Young Control vs. Old Polypharmacy	Compare the effects of young age without polypharmacy and polypharmacy in old age on MSCs.

**Table 2 ijms-25-05830-t002:** Excitation/emission channel values.

Channel	Peak Excitation (nm)	Peak Emission (nm)	Fluorophore
1	345	414	NADH/Elastin
2	345	451	NADH
3	345	575	Flavin/Lipo-Pigments
4	490	575	Flavin
5	505	575	Flavin
6	490	594	Flavin
7	505	594	Flavin
8	490	675	Porphyrin
9	505	675	Porphyrin
10	358	414	NADH/Elastin
11	371	414	NADH/Elastin
12	358	451	NADH/Elastin
13	377	451	NADH/Elastin
14	358	575	Flavin
15	381	575	Flavin
16	403	575	Flavin
17	430	575	Flavin
18	457	575	Flavin
19	358	594	Flavin/Lipo-pigments
20	377	594	Flavin/Lipo-pigments
21	381	594	Flavin
22	391	594	Flavin
23	400	594	Flavin
24	408	594	Flavin
25	437	594	Flavin
26	358	675	Porphyrin
27	381	675	Porphyrin
28	391	675	Porphyrin
29	437	675	Porphyrin
30	391	647	Flavin/Porphyrin
31	437	647	Flavin/Porphyrin
32	476	575	Brightfield Image

## Data Availability

Data will be made available on reasonable request.

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
