# Peer review of "Ageing and Polypharmacy in Mesenchymal Stromal Cells: Metabolic Impact Assessed by Hyperspectral Imaging of Autofluorescence"

_ijms, 2024, doi:10.3390/ijms25115830_

Round 1
Reviewer 1 Report
Comments and Suggestions for Authors
The manuscript entitled "Aging and Polypharmacy in Mesenchymal Stromal Cells:Metabolic Impact Assessed by Hyperspectral Imaging of Autofluorescence" is quite interesting and well-written. I like your tables regarding cell groups and study design and wavelengths of each fluorophore. Therefore, I suggest your manuscript to be published after a few text editing.
1) Lines 97 and 117 - I think reference is missing.
2) Give to table 1 and table 2 their names
Author Response
Dear Reviewers and Editors,
Thank you for your careful review and insightful comments. On reading the edited version of our manuscript it appears that a small error was introduced in the editorial process (not present in our submitted version) which caused one half of the references in the text to be formatted in the ‘Numbered’ style and the other half to be formatted in the ‘Author Date’ style. As only the Numbered references were displayed in the reference list, this gave the impression that our consideration of the literature was relatively light (23 references indicated as cited instead of the 36 that were), and that a higher proportion of our biography were auto citations than is the case.
We have corrected this issue in our re-uploaded manuscript in addition to addressing your thoughts and requests as detailed below.
Reviewer 1:
The manuscript entitled "Aging and Polypharmacy in Mesenchymal Stromal Cells:Metabolic Impact Assessed by Hyperspectral Imaging of Autofluorescence" is quite interesting and well-written. I like your tables regarding cell groups and study design and wavelengths of each fluorophore. Therefore, I suggest your manuscript to be published after a few text editing.
- Lines 97 and 117 - I think reference is missing.
Although the error is not present in our version we believe it is resulting from text ‘tags’ used in connection with figure legends. These have been removed and replaced with plain text, which should resolve the issue.
2) Give to table 1 and table 2 their names
Table 1 is now titled ‘Cell Group Comparisons’
Table 2 is now titled ‘Excitation/Emission Channel Values’
Reviewer 2 Report
Comments and Suggestions for Authors
Journal: International Journal of Mechanical Sciences
Ms ID: ijms-3000154
Title: Ageing and Polypharmacy in Mesenchymal Stromal Cells: Metabolic Impact Assessed by Hyperspectral Imaging of Autofluorescence
This article aimed to study the polypharmacy regimen may have a more profound impact on MSCs than ageing, and can simultaneously reduce optical redox ratio (ORR) and increase porphyrin levels. There have some issues need carefully addressed and revised as list below: 
Comments:
1. Please provide the reference citation in the “Introduction section” of line 57-65.
2. (Error! Reference source not found.1g) (line 97). Please explain.
2. Error! Reference source not found.3c (line 117). Please explain.
3. Please provide the description of “Figure 5A-5C, Figure 5e-5f) in the “Results” section .
4. Please provide the description of “Figure 6e-6f) in the “Results’ section.
5. Please provide more detail description in the “Results” section following list below: Are the age of mice and passage of MSC cells used in each experiment the same?
1) 2.1. Cells from Young and Old Control Mice Cells from Young and Old Control Mice: Age of mice? Passage of MSC cells?
1) 2.1. Cells from Young and Old Control Mice
2) 2.2. Cells from Young Polypharmacy and Old Polypharmacy Treated Mice
3) 2.3. Cells from Young Control and Young Polypharmacy Treated Mice
4) 2.4. Cells from Old Control and Old Polypharmacy Treated Mice
5) Cells from Young Polypharmacy and Old Control Treated Mice
6) Cells from Young Control and Old Polypharmacy Treated Mice
7.Please provide the MSC marker identification data for CD105, CD29, Sca-1, CD45 in this article.
8. Please provide more detail description as well as citation references how this article can imply for the use of autologous MSCs for older patients with chronic disease in the “Discussion” section.
9. Please provide the “Conclusion” section in this article.

none
Author Response
Dear Reviewers and Editors,
Thank you for your careful review and insightful comments. On reading the edited version of our manuscript it appears that a small error was introduced in the editorial process (not present in our submitted version) which caused one half of the references in the text to be formatted in the ‘Numbered’ style and the other half to be formatted in the ‘Author Date’ style. As only the Numbered references were displayed in the reference list, this gave the impression that our consideration of the literature was relatively light (23 references indicated as cited instead of the 36 that were), and that a higher proportion of our biography were auto citations than is the case.
We have corrected this issue in our re-uploaded manuscript in addition to addressing your thoughts and requests as detailed below.
Reviewer 2:
- Please provide the reference citation in the “Introduction section” of line 57-65.
Citations have been added in support of each of the assertions in this paragraph.
- (Error! Reference source not found.1g) (line 97). Please explain.
- Error! Reference source not found.3c (line 117). Please explain.
This error seems to have resulted from the use of a text ‘tag’. The tag has been removed and the plain text directing the reader to the relevant figures has been inserted.
- Please provide the description of “Figure 5A-5C, Figure 5e-5f) in the “Results” section .
The results section has been updated to correctly direct readers attention to the relevant panels of the figures are requested.
“The porphyrin and redox channels were successful in separating old control (OC) and YP cells along their respective axes (Figure 5d). YP cells had higher mean porphyrin, and lower ORR (Figure 5 a-c); this allowed for secondary channels to be created using the product of porphyrin channels, and ORR to act as training data for LRA. Thus, the subsequent model created by logistic regression analysis (LRA) (Figure 5f) was accurate in its classifications of YP and OC cells based on significant primary and secondary channels. Area under the ROC curve was 0.94 (Figure 5g) highlighting the accuracy at which the training data can produce a model that correctly categorises data.”
- Please provide the description of “Figure 6e-6f) in the “Results’ section.
As above, the Results section text related to Figure 6 has been updated to indicate which sections of Figure 6 need to be consulted for the support of each statement made. It now reads as follows:
“Separation between YC and OP cells was achieved using porphyrin, redox, and NAD(P)H channels, with PCA (Figure 6e), LRA (Figure 6f), and simple 3D plots (Figure 6d) all showing clear separation (Figure 6 g; ROC AUC=0.99). In cells from old mice re-ceiving polypharmacy, boxplots show significant decreases in NAD(P)H levels (Figure 6a) and ORR (Figure 6c) and increases in porphyrin levels (Figure 6b).”
- Please provide more detail description in the “Results” section following list below: Are the age of mice and passage of MSC cells used in each experiment the same?
1) 2.1. Cells from Young and Old Control Mice Cells from Young and Old Control Mice: Age of mice? Passage of MSC cells?
1) 2.1. Cells from Young and Old Control Mice
2) 2.2. Cells from Young Polypharmacy and Old Polypharmacy Treated Mice
3) 2.3. Cells from Young Control and Young Polypharmacy Treated Mice
4) 2.4. Cells from Old Control and Old Polypharmacy Treated Mice
5) Cells from Young Polypharmacy and Old Control Treated Mice
6) Cells from Young Control and Old Polypharmacy Treated Mice
Yes, all cells were at the same passage (passage 2) and all mice were the same age (within the Old Young age groupings). The below text has been inserted at the start of the results section to better inform readers of this context rather than requiring them to read to the Methods section:
“All experiments were performed on MSCs that had been cultured to passage 2. Old mice were aged 25.5 months at the time of sacrifice and cell line derivation, while young mice were aged 6.5 months.”
7.Please provide the MSC marker identification data for CD105, CD29, Sca-1, CD45 in this article.
The details of the cell surface marker validation are now included at the beginning of the results section along with the new text describing the mouse ages.
- Please provide more detail description as well as citation references how this article can imply for the use of autologous MSCs for older patients with chronic disease in the “Discussion” section.
We have addressed the relevance of our findings to this group in the following paragraph which has been added to the discussion:
“Our findings support that the biochemistry of MSC lines is influenced by the status and exposures experienced by their original donor. As MSC lines used for autologous therapies will – with the exception of pre-banked cells – invariably come from individuals experiencing at least one pathology, and with a high likelihood of polypharmacy and advanced age, this warrants serious consideration for the design of studies and the implementation of therapies. In particular, it may be a contributing factor to the generally heterogenous nature and behaviour of human MSC lines [1] as well as difficulties in translating the findings of pre-clinical studies to patient outcomes [2, 3]. Further studies demonstrating effects on clinical outcomes are needed to make any recommendations, however de-prescribing – the careful and considered withdrawal of medications whose benefits may no longer outweigh their negative effects – is already good clinical practice [4]. This would represent a new angle of research for ‘pre’-habilitation – an emerging health care strategy, including in preparation for stem cell therapies [5, 6] – where efforts are made to improve the patient’s fitness so that they can better bear and recover from interventions (e.g. chemotherapy or surgery). In the proposed scenario, however, the patient’s health would be improved with the goal of improving the quality of the therapy itself. In addition to deprescribing, interventions like nutritional supplementation with the metabolic precursor NMN – which has been shown to be able to improve the quality of subsequently derived MSC lines [7] – may also be able to improve the quality of MSC lines from older people exposed to polypharmacy undergoing autologous stem cell therapy.”
- Please provide the “Conclusion” section in this article.
The existing conclusion has been demarcated with the requested subheading.
Reviewer 3 Report
Comments and Suggestions for Authors
Review_ijms-3000154
The aim of the study is to examine the impact of the polypharmacy on the MSCs features and their usefulness in the potential cell therapy. The publication is interesting, concise and well-written, with has correct assumptions and properly described methodology. However, in my opinion, the manuscript contains a too far-reaching conclusions according to obtained results and material and method used (e.g. low number of experimental animals in each group).
Abstract is clear and contains goal of the study and main results. Introduction is short and clear, goal of the study and hypothesis were provided. The Material and Method section contains some uncertainties and divergencies (the list is provided below). The Results are interesting, but this section has to be revised and completed with accurate reference sources. Discussion is concise and interesting. The list of literature is uncompleted – provided citations are present only in the Introduction, the many of them were recognized as auto-citations. The format of the citation in Discussion section is different than in the rest of the manuscript. These citations are not provided in the Literature section.
Results
L 97, 117 – Error?? Please provide data, which were described in this section or remove statements, which are not supported by the charts
Fig.1, 3, 4,5, 6 – there is no information about the statistical tests used in the figure description
Material & Methods
In materials & methods section there is a number of uncertainties and discrepancies, including regarding the method of cell analysis. Please find below detailed doubts:
- what was the age of the animals? The authors say in the introduction, that it was 2,5 months + 2,5 months treatment for young mice, and 21,5+2,5 months for old ones, which means, that MSCs should be collected in the age of 5 and 24 months for young and old mice, respectively. In this section 6,5 and 25,5 months were given – please explain this divergency
- 4 mice in the group seems not to be enough for getting reliable statistical data
- What does it “A convenience sample was used from the cohort to conduct the MSC experiments” mean? Were cells obtained from different animals pooled and analyzed together? If yes, what is the reason? What about aspect of individual variability?
- L307-312 – the most of provided citations in this section are not present in the literature section
Literature
The authors cite 23 publications, half of which were referred to in the first paragraph of the manuscript. Most of them are reviews, focusing on the definition of polypharmacy and the technique of testing using autofluorescence. Moreover, the literature contains many self-citations – although this area of research is interesting, but it does not seem so niche that it would be advisable to cite one's own works to a large extent. Therefore, the review of the cited literature seems to be insufficient and does not explain the significance of the obtained results. Please complete this section with the positions cited in the Discussion.
The minor mistakes:
L48 – unnecessary underscore
L78 – please provide explanation of hMSCs shortcut
L93 – ‘compared you’?
L89, 270, 285, 307, 166 – please delete an extra space
L307 - please put the space before the bracket
Tables’ titles need to be completed

Author Response
Dear Reviewers and Editors,
Thank you for your careful review and insightful comments. On reading the edited version of our manuscript it appears that a small error was introduced in the editorial process (not present in our submitted version) which caused one half of the references in the text to be formatted in the ‘Numbered’ style and the other half to be formatted in the ‘Author Date’ style. As only the Numbered references were displayed in the reference list, this gave the impression that our consideration of the literature was relatively light (23 references indicated as cited instead of the 36 that were), and that a higher proportion of our biography were auto citations than is the case.
We have corrected this issue in our re-uploaded manuscript in addition to addressing your thoughts and requests as detailed below.
Reviewer 3:
The aim of the study is to examine the impact of the polypharmacy on the MSCs features and their usefulness in the potential cell therapy. The publication is interesting, concise and well-written, with has correct assumptions and properly described methodology. However, in my opinion, the manuscript contains a too far-reaching conclusions according to obtained results and material and method used (e.g. low number of experimental animals in each group).
Abstract is clear and contains goal of the study and main results. Introduction is short and clear, goal of the study and hypothesis were provided. The Material and Method section contains some uncertainties and divergencies (the list is provided below). The Results are interesting, but this section has to be revised and completed with accurate reference sources. Discussion is concise and interesting. The list of literature is uncompleted – provided citations are present only in the Introduction, the many of them were recognized as auto-citations. The format of the citation in Discussion section is different than in the rest of the manuscript. These citations are not provided in the Literature section.
Thank you for your review and commentary. We have addressed all of your specific issues as detailed below. The issue with the literature/citations has been resolved as described at the head of the letter.
Results
L 97, 117 – Error?? Please provide data, which were described in this section or remove statements, which are not supported by the charts
These error messages were mis formatted figure tags. Now that they have been removed and replaced with plain text they direct readers to the appropriate figure sections for the statements made.
Fig.1, 3, 4,5, 6 – there is no information about the statistical tests used in the figure description
All figure descriptions have been updated to include statistical details of the tests used in the represented data.
Material & Methods
In materials & methods section there is a number of uncertainties and discrepancies, including regarding the method of cell analysis. Please find below detailed doubts:
- what was the age of the animals? The authors say in the introduction, that it was 2,5 months + 2,5 months treatment for young mice, and 21,5+2,5 months for old ones, which means, that MSCs should be collected in the age of 5 and 24 months for young and old mice, respectively. In this section 6,5 and 25,5 months were given – please explain this divergency
Thank you for highlighting this discrepancy. The correct age of the mice are the 6.5/25.5 values. The text has not been updated to ensure consistency and accuracy.
- 4 mice in the group seems not to be enough for getting reliable statistical data
Analyses have been undertaken and reported at the single cell level and the power of our statistical testing is therefore not a function of included animals which contextually represent biological replicates as opposed to N-values.
- What does it “A convenience sample was used from the cohort to conduct the MSC experiments” mean? Were cells obtained from different animals pooled and analyzed together? If yes, what is the reason? What about aspect of individual variability?
We apologise for the unclear wording, we have rephrased the relevant section and also added details pertaining to your additional queries as follows:
“At the point of analysis, after attrition due to premature mortality amongst the aged mice and unsuccessful line derivation, the groups contained 6, 6, 6 and 4 mice for young control, old control, young polypharmacy and old polypharmacy, respectively. Data within groups was pooled and analysed at the single cell level as detailed in section 4.4.”
- L307-312 – the most of provided citations in this section are not present in the literature section
The formatting error in the Endnote library has been corrected and all citations are now correctly represented.
Literature
The authors cite 23 publications, half of which were referred to in the first paragraph of the manuscript. Most of them are reviews, focusing on the definition of polypharmacy and the technique of testing using autofluorescence. Moreover, the literature contains many self-citations – although this area of research is interesting, but it does not seem so niche that it would be advisable to cite one's own works to a large extent. Therefore, the review of the cited literature seems to be insufficient and does not explain the significance of the obtained results. Please complete this section with the positions cited in the Discussion.
Due to the formatting error in the bibliography, only the references cited in the Introduction section of the text were shown in the References section. This gave the incorrect impression that only 23 citations were referenced when it was actually higher. Moreover, the Introduction hewed towards explaining the rationale of our study as well as the underpinning technology, which heavily weighted it towards self citations, however with the references included in the Discussion included in the calculus, our review of the literature was broader than it appeared. It has been further expended in addressing reviewer comments (now 48) which also majorly reduces to portion of self citations. We believe that the quality of the literature review is now much improved from the version first submitted and thank the reviewer’s for their direction.
The minor mistakes:
L48 – unnecessary underscore
Underscore removed.
L78 – please provide explanation of hMSCs shortcut
As this was the only iteration of the abbreviation, we have provided the wording in full (human MSCs)
L93 – ‘compared you’?
“you” has been corrected to “to”
L89, 270, 285, 307, 166 – please delete an extra space
Extra spaces have been deleted.
L307 - please put the space before the bracket
The space has been inserted.
Tables’ titles need to be completed
Titles are now provided.
Round 2
Reviewer 2 Report
Comments and Suggestions for Authors
Authors were adequately responded all comments. However, there remain a question need addressed before accept for this article.
Comment:
Please revised " in<3% of cells" (line 96) into "in <3% of cells (data not shown).
Comments on the Quality of English Language
none
Author Response
The change has been made as requested.